# Cross-species analysis of LZTR1 loss-of-function mutants demonstrates dependency to RIT1 orthologs

Antonio Cuevas-Navarro[1], Laura Rodriguez-Muñoz[2], Joaquim Grego-Bessa[3], Alice Cheng[1], Katherine A Rauen[4,5], Anatoly Urisman[6], Frank McCormick[1], Gerardo Jimenez[2,7], Pau Castel[8]*

[1]Helen Diller Family Comprehensive Cancer Center, University of California, San Francisco, San Francisco, United States; [2]Institute for Molecular Biology of Barcelona, Consejo Superior de Investigaciones Científicas, Barcelona, Spain; [3]Centro Nacional de Investigaciones Cardiovasculares, Madrid, Spain; [4]UC Davis MIND Institute, University of California Davis, Sacramento, United States; [5]Department of Pediatrics, University of California Davis, Sacramento, United States; [6]Department of Anatomic Pathology, University of California San Francisco, San Francisco, United States; [7]Institució Catalana de Recerca i Estudis Avançats (ICREA), Barcelona, Spain; [8]Department of Biochemistry and Molecular Pharmacology, New York University Grossman School of Medicine, New York, United States

**Abstract** RAS GTPases are highly conserved proteins involved in the regulation of mitogenic signaling. We have previously described a novel Cullin 3 RING E3 ubiquitin ligase complex formed by the substrate adaptor protein LZTR1 that binds, ubiquitinates, and promotes proteasomal degradation of the RAS GTPase RIT1. In addition, others have described that this complex is also responsible for the ubiquitination of classical RAS GTPases. Here, we have analyzed the phenotypes of *Lztr1* loss-of-function mutants in both fruit flies and mice and have demonstrated a biochemical preference for their RIT1 orthologs. Moreover, we show that *Lztr1* is haplosufficient in mice and that embryonic lethality of the homozygous null allele can be rescued by deletion of *Rit1*. Overall, our results indicate that, in model organisms, RIT1 orthologs are the preferred substrates of LZTR1.

*For correspondence:
pau.castel@nyulangone.org

## Editor's evaluation

Using elegant cross-species biochemistry and genetic approaches, this paper describes the role of the ubiquitin adaptor protein LZTR1 in regulation of the RAS-related GTPase RIT1 as its principal substrate involved in the RASopathy, Noonan syndrome. Although this work does not fully rule out the involvement of canonical RAS isoforms in LZTR1-associated RASopathies in humans, the extensive genetic experiments in *Drosophila* and mouse presented here suggest that pathological phenotypes observed in LZTR1-linked RASopathy models are mediated primarily by its target RIT1, and not by canonical RAS isoforms.

## Introduction

Dysregulation of signal transduction by the RAS family of guanosine 5'-triphosphate (GTP) hydrolases (GTPases) can have profound effects on human development and cause genetic disorders collectively termed RASopathies (*Castel et al., 2020*; *Rauen, 2013*). Ras GTPases exhibit high affinity toward

guanine nucleotides and act as molecular switches by mediating GTP hydrolysis. The nucleotide cycling of RAS GTPases is tightly regulated by GTPase activating proteins (GAPs; e.g. neurofibromin) and guanine nucleotide exchange factors (GEFs; e.g. SOS1) that facilitate nucleotide hydrolysis or loading, respectively (*Simanshu et al., 2017*). Upon GTP binding, RAS proteins undergo a conformational change, which promotes the interaction with different protein effectors that activate downstream signaling pathways, including Raf/MEK/ERK mitogen activated protein kinase (MAPK) and phosphoinositide 3-kinase (PI3K) pathways. Although GAPs and GEFs can rapidly affect the nucleotide cycling of RAS proteins, and hence their activity, other accessory proteins can modulate downstream signaling by regulating the stability and/or activity of RAS GTPases.

Noonan syndrome (NS) is a common RASopathy that is characterized by craniofacial dysmorphism, short stature, congenital heart disease, and developmental delays. The cause of NS has been linked to genetic alterations that result in hyperactivation of the Raf/MEK/ERK MAPK pathway, including recently reported gain-of-function mutations in the RAS GTPase RIT1 and loss-of-function mutations in the BTB protein LZTR1 (*Aoki et al., 2013*; *Yamamoto et al., 2015*; *Johnston et al., 2018*). We have previously reported that RIT1 is mostly bound to GTP in cells, suggesting that it either lacks a GAP, uses a less active GAP, or relies on its intrinsic GTPase activity (*Castel et al., 2019*). An alternative regulatory mechanism of RIT1 activity is through protein degradation; we identified a Cullin-3 RING E3 ubiquitin ligase complex (CRL3) that uses LZTR1 as a substrate receptor (CRL3$^{LZTR1}$) to bind RIT1 and promote its ubiquitination and proteasomal degradation. CRL3$^{LZTR1}$ binds to GDP-bound RIT1 and maintains tight regulation of RIT1 protein levels. Importantly, NS-associated *RIT1* and *LZTR1* missense mutations disrupt CRL3$^{LZTR1}$-RIT1 binding, thus relieving RIT1 from its negative regulatory mechanism (*Castel, 2022*).

LZTR1 has been reported to regulate the protein stability of other RAS GTPases by similar mechanisms (*Bigenzahn et al., 2018*; *Steklov et al., 2018*; *Abe et al., 2020*). In this study, we employed a molecular evolutionary approach to elucidate the functional relationship and co-evolution of LZTR1 and its cognate RAS GTPase substrate(s), including the evaluation of LZTR1 loss-of-function mutations in invertebrate and mammalian model organisms.

## Results and discussion

The RAS family of GTPase proteins has been previously shown to be well-conserved across species and LZTR1 has been proposed to regulate a subset of these proteins (*Rojas et al., 2012*). Therefore, to understand how LZTR1 has evolved in comparison with other RAS GTPases, we performed protein alignment analysis of the human orthologs of KRAS, RIT1, and LZTR1 orthologs from human and common model organisms. Consistent with previous studies, KRAS was highly conserved in less complex organisms, including fission yeast (Ras1; 60% protein identity) (*Figure 1a*). In contrast, we found that RIT1 rapidly diverged in less complex organisms and a mildly conserved protein in *Drosophila melanogaster* (RIC; 60% protein identity) was the most distant ortholog that could be found in these laboratory model species. LZTR1 followed a similar pattern as RIT1; we could identify an ortholog in fruit flies (CG3711/LZTR1; 60% protein identity), but not in the roundworm *Caenorhabditis elegans* (*Figure 1a*). These analyses suggest that RIT1 and LZTR1 are more likely to have functionally co-evolved than KRAS and LZTR1. To further assess the biochemical relationship between these proteins across model organisms, we generated recombinant proteins for the KRAS, RIT1, and LZTR1 orthologs from mouse (*Mus musculus*), zebrafish (*Danio rerio*), and fruit fly (*Drosophila melanogaster*) and tested their ability to interact in pull down assays. All LZTR1 orthologs preferentially interacted with RIT1 orthologs (*Figure 1b*, *Figure 1—figure supplement 1*), indicating that this interaction that we previously reported for human proteins is also conserved in less complex model organisms.

To better understand the role of the LZTR1 ortholog in *Drosophila melanogaster*, we isolated *Lztr1* loss-of-function mutations by using two independent CRISPR/Cas9 gene editing approaches. One resulting allele, *Lztr1[1]*, causes a frameshift at the beginning of the second exon and encodes a short, truncated product. Another allele, *Lztr1[2]*, lacks all coding sequences and is, therefore, a null mutation (*Figure 2a*). *Lztr1* null flies were normal and fertile and did not display any obvious phenotype. A previous study showed that flies expressing RNA interference constructs against *Lztr1* displayed minor defects in wing vein patterning (*Bigenzahn et al., 2018*); however, we did not find consistent vein defects in our null mutants (*Figure 2b*).

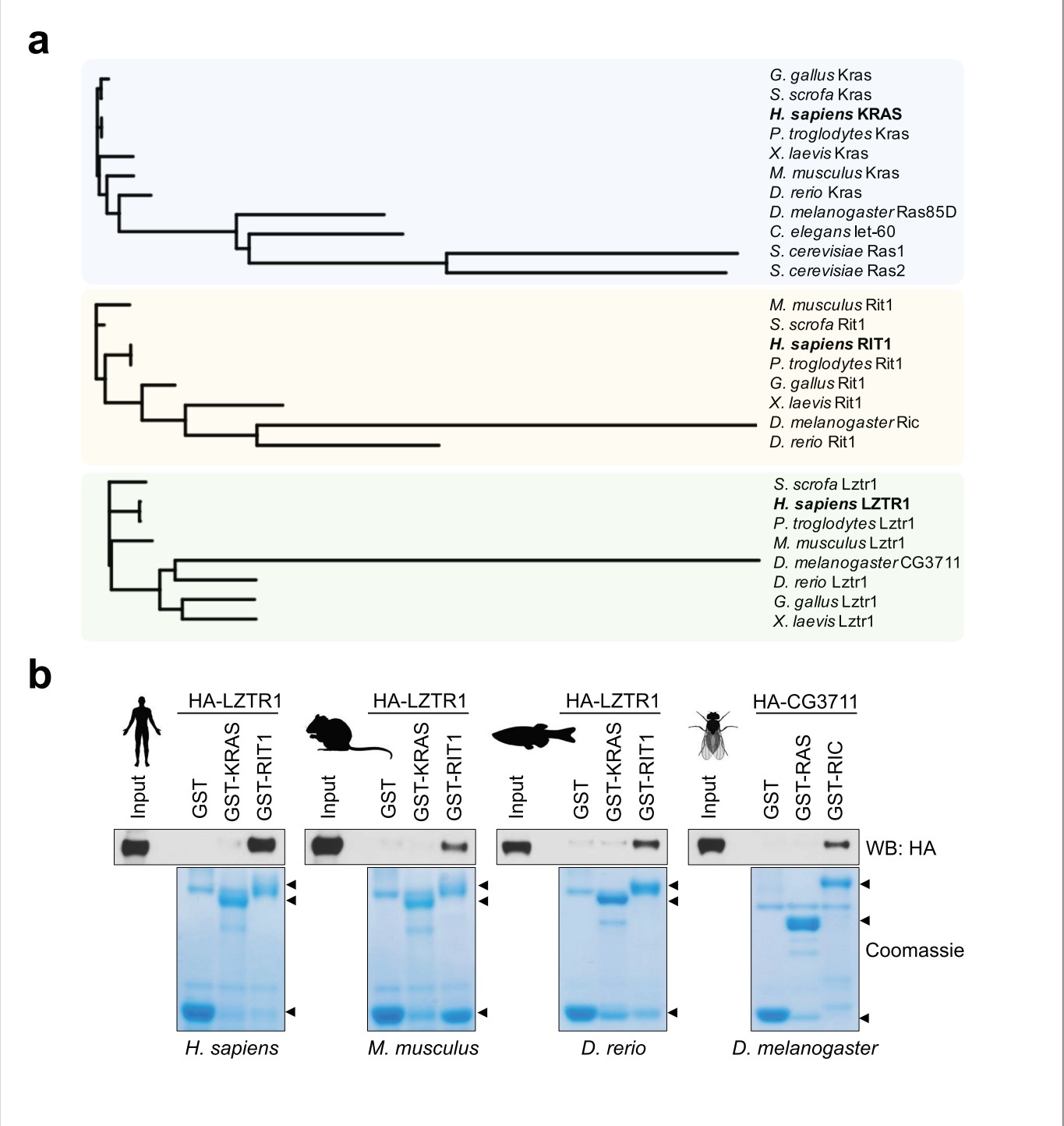

**Figure 1.** *Evolutionary analysis of LZTR1 and the RAS proteins KRAS and RIT1.* (**a**) Phylogenetic trees of KRAS, RIT1, and LZTR1 orthologs based on multiple protein sequence alignments performed with Clustal Omega (*Figure 1—source data 1*). Orthologs were searched in the following model organisms: chimpanzee (*Pan troglodytes*), pig (*Sus scrofa*), chicken (*Gallus gallus*), mouse (*Mus musculus*), African clawed frog (*Xenopus laevis*), zebrafish (*Danio rerio*), fruit fly (*Drosophila melanogaster*), roundworm (*Caenorhabditis elegans*), and budding yeast (*Saccharomyces cerevisiae*). (**b**) Pull-down assays using GST-tagged KRAS and RIT1 or their mouse, zebrafish, or fruit fly orthologs produced recombinantly in bacteria. Recombinant proteins were incubated with lysates from HEK-293T cells expressing their corresponding species' HA-tagged LZTR1 ortholog. Representative results from three biological replicates.

The online version of this article includes the following source data and figure supplement(s) for figure 1:

**Source data 1.** Raw data for *Figure 1*.

*Figure 1 continued on next page*

*Figure 1 continued*

**Figure supplement 1.** Interaction between the human SOS^cat domain and the different KRAS orthologs using pull-down assays demonstrates that the proteins are properly folded and GDP loaded in our assay.

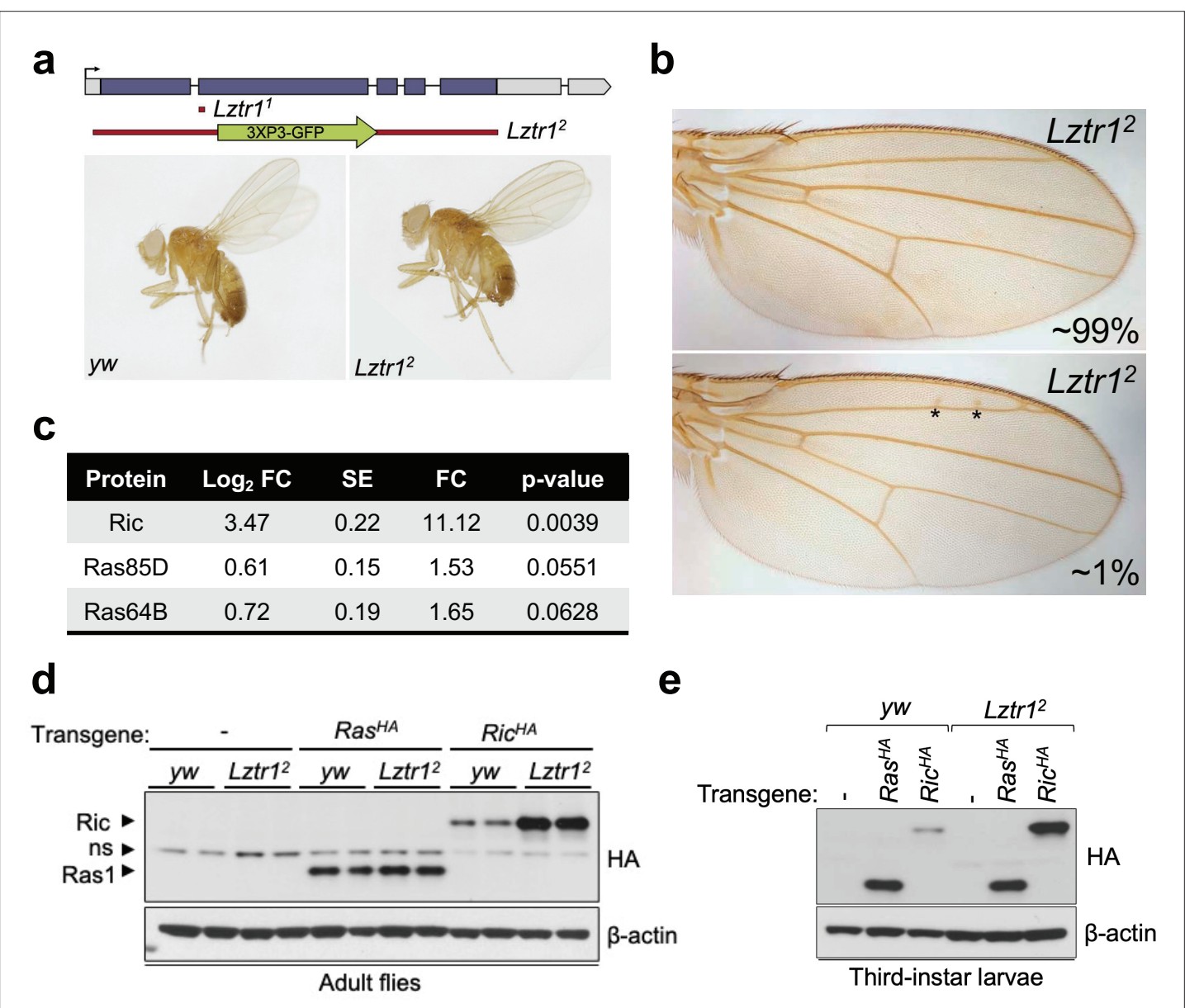

**Figure 2.** *Drosophila* Lztr1 regulates Ric stability. (**a**) A schematic representation of the *Drosophila CG3711/Lztr1* gene locus is shown. Coding exons are represented in blue. Two CRISPR/Cas9-mediated approaches were used to isolate the *Lztr1* loss-of-function alleles, *Lztr1^1* and *Lztr1^2*. (**b**) Wing vein patterning is not affected in *Lztr1* null flies (upper panel), with very few individuals exhibiting small ectopic veinlets in two or three points (asterisks; lower panel) (n = 209 flies). (**c**) Estimated normalized protein abundance expressed as mean log2 fold change in *Lztr1^2* vs *yw* (control) comparison. Corresponding single-protein standard error (SE) and t-test p-values are listed. (**d**) Immunoblot analysis of protein extracts isolated from the indicated transgenic adult flies in a *yw* or *Lztr1^2* background. ns: non-specific band. (**e**) Same as panel (**d**) with protein extracts isolated from third-instar larvae. Representative results from three biological replicates.

The online version of this article includes the following source data for figure 2:

**Source data 1.** Raw data for *Figure 2*.

Next, we undertook label-free proteomics to quantify the levels of Ras family proteins in head extracts from *Lztr1²* and background matched control (*yw*) flies. Ric levels were upregulated ~11 times in *Lztr1²* fly heads relative to the control, while Ras85D (the HRAS, KRAS, and NRAS ortholog, hereafter referred to as Ras) levels were only modestly altered in both backgrounds (~1.5-fold difference) (*Figure 2c*). Similarly, Ras64D (the MRAS ortholog) was only upregulated ~1.6-fold in *Lztr1²* mutant heads. To validate these results, we generated *Drosophila* transgenic lines carrying N-terminally HA tagged Ric or Ras genes under the control of their natural genomic sequences, thereby facilitating detection of both products at their endogenous levels. Both transgenes were then placed in the *Lztr1²* mutant background to assess the effect of *Lztr1* depletion on Ric and Ras protein stability. Consistent with our mass spectrometry results, HA-Ric levels were elevated in the absence of Lztr1, while HA-Ras levels remained barely altered in either mutant adult flies or third-instar larvae (*Figure 2d–e*). Altogether, these experiments show that *Drosophila* Lztr1 preferentially regulates Ric rather than Ras levels, as seen with the corresponding mammalian orthologs.

Given the absence of a phenotype in fruit flies, we examined the effect of *Lztr1* deletion in mice. Steklov et al had previously reported that *Lztr1* heterozygous knockout mice display typical features of NS, suggestive of haploinsufficiency (*Steklov et al., 2018*). In humans, NS-associated variants are most commonly found as bi-allelic loss-of-function variants that are inherited in an autosomal recessive manner; although there are few heterozygous variants that segregate as autosomal dominant, these are single nucleotide variants that likely act as dominant negatives (*Johnston et al., 2018*; *Yamamoto et al., 2015*; *Motta et al., 2019*). Hence, we further analyzed the potential NS-like phenotype of heterozygous *Lztr1* deletion in the mouse as previously described in other mouse models (*Araki et al., 2004*; *Wu et al., 2011*; *Hernández-Porras et al., 2014*; *Castel et al., 2019*). Heterozygous mice did not exhibit significant changes in body weight and, overall, their morphology appeared normal (*Figure 3a*, *Figure 3—figure supplement 1*). The morphology of the skull, which generally shows dysmorphic features in mouse models of NS (round skull, blunt snout, and hypertelorism), was assessed using micro-computed tomography (μCT) and did not have significant differences when compared to wild-type littermates (*Figure 3b*). In addition, we assessed whether heterozygous mice displayed either cardiomegaly or splenomegaly, as these signs have been observed in other NS-like mouse models, but we failed to observe differences with wild-type animals (*Figure 3c–d*). No significant changes in RAS GTPases levels were seen either in tissue extracts from these mice (*Figure 3—figure supplement 1*). These results indicate that *Lztr1* heterozygous mice do not exhibit the typical features seen in other NS mouse models and indicate that *Lztr1* is haplosufficient for this phenotype. In support of these observations in mice, NS families carrying *LZTR1* alleles generally exhibit an autosomal recessive inheritance pattern and bi-allelic inactivation of *LZTR1* is required to exhibit a NS phenotype (*Johnston et al., 2018*). In addition, analysis of LZTR1 variants in the genome aggregation database (gnomAD) shows that many loss-of-function variants are observed in healthy individuals (pLI = 0; o/e = 2.28) (*Karczewski et al., 2020*). This indicates that these heterozygous loss-of-function variants are tolerated and are found in non-syndromic individuals. Thus, these data suggest haploinsufficiency of *LZTR1* in humans is unlikely to lead to NS.

We and others have previously shown that *Lztr1* knockout mice are embryonically lethal (*Steklov et al., 2018*; *Castel et al., 2019*); however, the reason for such lethality remains largely unknown. A previous study showed that *Lztr1*-associated embryonic lethality can be rescued by administering a MEK1/2 inhibitor to pregnant females, indicating that the phenotype is largely dependent on MAPK hyperactivation (*Steklov et al., 2018*). In fact, in other murine NS models, excessive activation of MAPK during embryonic development results in lethality, as seen in the *Kras*^V14I model (*Hernández-Porras et al., 2014*). Interestingly, this phenotype appears to be more prominent in certain mouse background strains, such as C57BL/6 N, and more permissive in 129Sv (*Araki et al., 2004*; *Hernández-Porras et al., 2015*). Since our *Lztr1* mutant mice are in the C57BL/6 N strain, we hypothesized that mouse background affects embryonic lethality. Backcrossing our mice to 129Sv females yielded heterozygous *Lztr1* mutant mice in a 50% mixed background. After one backcross (referred to as F1), using a heterozygous x heterozygous breeding scheme, we obtained 1/55 (~1.8%) homozygous viable mice. At F2, we obtained 3/45 (~6.7%) homozygous viable mice. At F3, we obtained 3/34 (~8.8%) homozygous viable mice (*Figure 3e*). In contrast, mice in the 129Sv pure background did not yield any homozygous viable mice, suggesting that a combination of strain-specific genetic modifiers is likely required to tolerate *Lztr1* deletion. In *Raf1* D486N NS mice, a potential gene modifier was mapped at

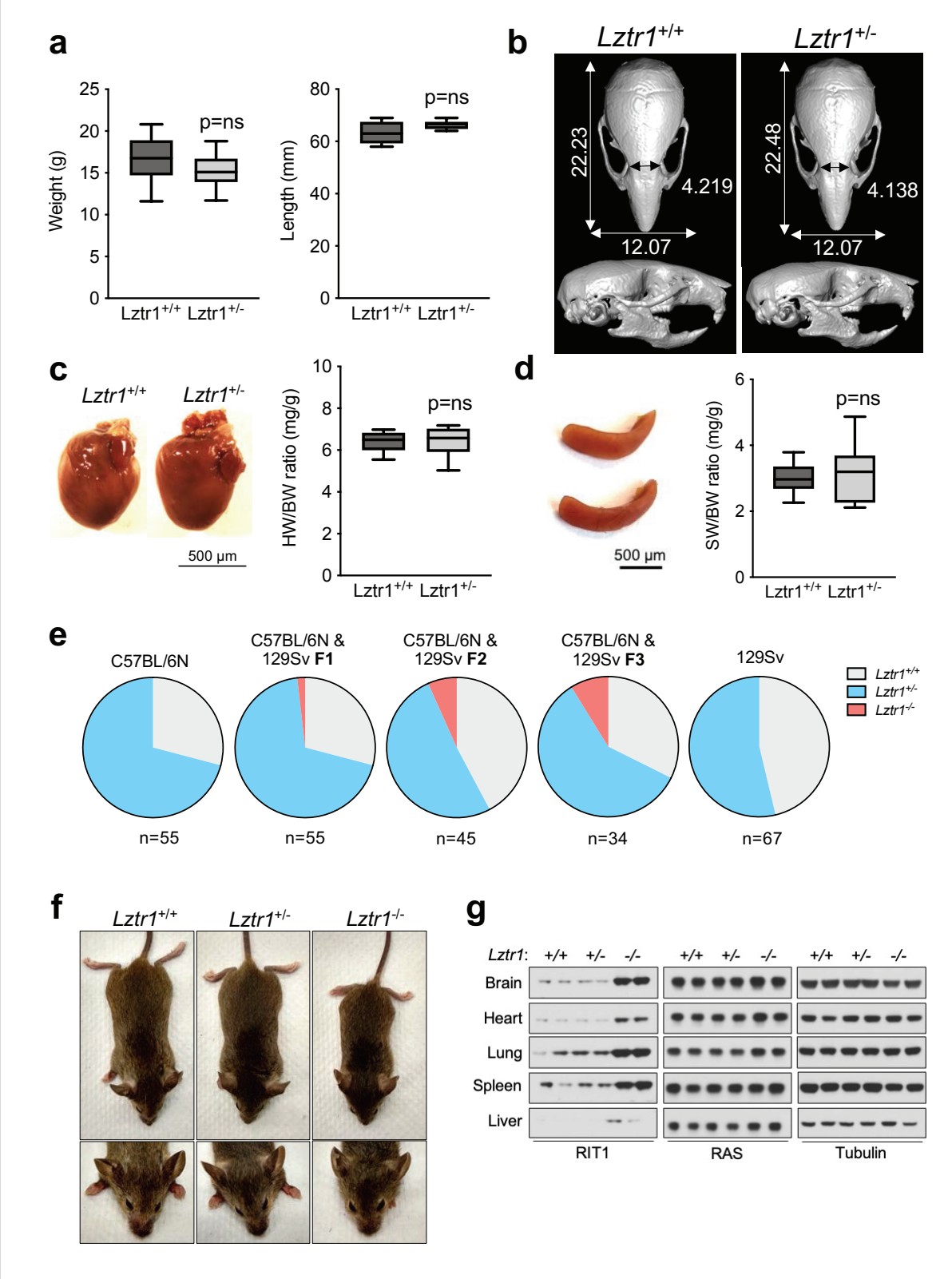

**Figure 3.** *Lztr1* is haplosufficient in mice and its null phenotype can be modified by strain background. (**a**) Weight (left panel) and length (right panel) of 4-week-old male *Lztr1* wild type (n = 18) and heterozygous mutant (n = 20) mice. (**b**) Representative µCT imaging of the skull of an 8-week-old male *Lztr1* wild type and heterozygous mutant mouse. The indicated values show the average measurement (mm) of length, width, and inner intercanthal distance in *Lztr1* wild type (n = 5) and heterozygous mutant (n = 5) mice. Mann-Whitney p values were not significant for all the measurements. (**c**) Heart

*Figure 3 continued*

weight was similar between 8-week-old male *Lztr1* wild type (n = 8) and heterozygous mutant (n = 6) mice, as assessed by heart to body weight ratio (HW/BW). Mann-Whitney test p value was not significant. (**d**) Same as panel (**c**), for the spleen of these mice. (**e**) Pie charts indicate the percentage of obtained genotypes upon weaning (21 days of age) the offspring of *Lztr1* heterozygous mutant intercrosses. Each pie chart represents a different strain background and/or mixed background. (**f**) Representative image of female littermates with the indicated *Lztr1* genotypes (C57BL/6N-129Sv F3 background). Note the decreased size, round skull, and proptosis of the homozygous *Lztr1* knockout mouse. (**g**) Immunoblot analysis of RIT1, RAS, and Tubulin proteins isolated from the indicated tissues of *Lztr1* wild type, heterozygous, and homozygous mice (C57BL/6N-129Sv F3 background). Protein lysates from two different mice were used for each genotype.

The online version of this article includes the following source data and figure supplement(s) for figure 3:

**Source data 1.** Raw data for *Figure 3*.

**Figure supplement 1.** Phenotyping results of Lztr1 heterozygous knockout female mice.

chromosome 8 of the 129Sv strain (*Wu et al., 2012*); however, we hypothesize that in the context of *Lztr1*, a negative gene modifier is also likely to be located in the Y chromosome of the 129Sv strain, given that we do not yield any homozygous viable mice after Y-chromosome fixing.

The limited number of viable homozygous *Lztr1* knockout mice prevented us from undertaking quantitative phenotyping; however, when compared to wild type littermates, *Lztr1* knockout mice appeared smaller, displayed characteristic dysmorphic facial features (round snout, hypertelorism, low set of ears, and proptosis), and exhibited cardiomegaly, consistent with other mouse models of NS (*Figure 3f*). In addition, *Lztr1* knockout mice exhibited increased levels of Rit1 in all the tissues that we analyzed, while RAS levels remained mostly unchanged (*Figure 3g*).

Next, we sought to investigate the cause of embryonic lethality in *Lztr1* knockout mice. We and others had previously shown that many embryos survive to late developmental stages (i.e. embryonic day (E)17.5–19.5), therefore, we harvested embryos on day E18.5. Most embryos display extensive hemorrhages (*Figure 4a*) and, consistent with this observation, conditional deletion of *Lztr1* in blood vessels has been previously shown to cause vascular leakage (*Sewduth et al., 2020*). Since many NS alleles that cause lethality in mice have been related to cardiovascular dysfunction, we analyzed the heart phenotype of E18.5 embryos. Valve leaflets and endocardial cushions were normal, in contrast to the previous *Ptpn11* NS-associated alleles (*Araki et al., 2004*). In *Lztr1* knockout embryos, the ventricular myocardial wall showed defects that are compatible with ventricular noncompaction cardiomyopathy, similar to that seen in the *Ptpn11* Q79R transgenic mouse model (*Nakamura et al., 2007*). Detailed analysis of compact and trabecular myocardial thickness showed significant differences in mutant embryos (*Figure 4b*). In addition, we observed interventricular septal defects in ~20% mutant hearts. Although the heart phenotype is striking, it is likely that embryonic lethality results from a combination of vascular and cardiac defects during development.

*Lztr1* embryonic lethality provides a unique model to undertake a genetic epistatic rescue experiment to assess which RAS GTPase is the critical downstream substrate of the CRL3$^{LZTR1}$ complex. We have previously shown that at the biochemical level CRL3$^{LZTR1}$ complex preferentially binds and promotes degradation of RIT1. Therefore, we generated *Rit1* knockout mice to test whether Rit1 depletion can rescue *Lztr1*-mediated embryonic lethality. To avoid potential heterosis, we generated *Rit1* knockouts in the C57BL6/N background; mice were born at expected Mendelian ratios, were fertile, and did not exhibit any visible phenotype, similar to a previous *Rit1* null mouse strain (*Cai et al., 2011*). We confirmed the elimination of Rit1 by immunoblot in different tissues, including brain, lung, and heart (*Figure 4c*). Next, we crossed *Rit1* knockout mice with *Lztr1* heterozygous mice to obtain both *Lztr1$^{-/+}$/Rit1$^{+/+}$* and *Lztr1$^{-/+}$/Rit1$^{-/-}$* progeny. As previously shown, a heterozygous breeding scheme with *Lztr1$^{-/+}$/Rit1$^{+/+}$* mice did not yield any viable *Lztr1* knockout mice (*Figures 3e and 4d*; *Sewduth et al., 2020*). In contrast, when *Lztr1$^{-/+}$/Rit1$^{-/-}$* breeders were used, we obtained 13/85 (~15.3%) mice that were double knockout (DKO) for both *Lztr1* and *Rit1* (*Figure 4d*). This result shows that *Rit1* deletion can rescue the embryonic lethality caused by *Lztr1* deletion in mice, indicating that Rit1 is the critical substrate of the CRL3$^{LZTR1}$ complex during embryonic development. The resulting DKO mice appeared normal, were fertile, and were absent of any detectable phenotype that resembled other NS mouse models, as assessed by size, heart weight, and cranial morphology (*Figure 4—figure supplement 1*). In addition, *Lztr1/Rit1* DKO embryos harvested at E18.5 showed a complete rescue of the cardiac phenotype, which were indistinguishable from *Rit1* KO embryos (*Figure 4e*, *Figure 4—figure supplement 1e*). To assess whether the Rit1 dependency established by

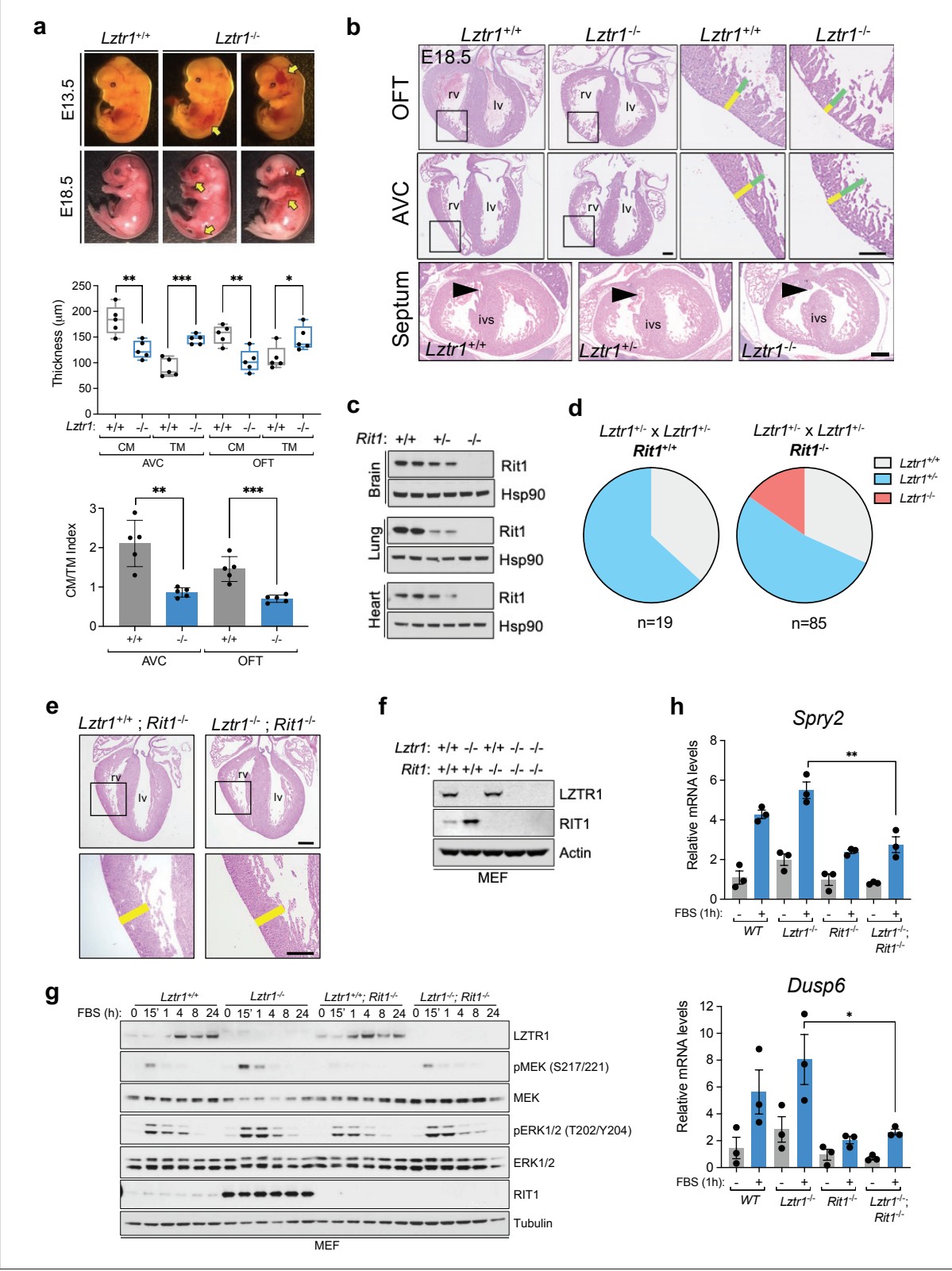

**Figure 4.** Lethality in Lztr1 knockout mutants as a result of Rit1-dependent cardiovascular defects. (**a**) Gross morphology of *Lztr1⁺/⁺* (wild type) and *Lztr1⁻/⁻* (knockout) embryos at E13.5 and E18.5. Note the presence of extensive hemorrhage in knockout embryos (yellow arrows). (**b**) Histological characterization of E18.5 hearts shows highly penetrant defects in ventricular wall thickness at both outflow tract (OFT) and atrioventricular canal (AVC) levels, as well as septal defects (arrows) in ~20% *Lztr1⁻/⁻* embryos. Quantification of ventricular wall thickness in both CM (yellow) and TM (green)

*Figure 4 continued on next page*

*Figure 4 continued*

and CM/TM index are shown in the graphs (n = 5). Rv: right ventricle; lv: left ventricle; ivs: interventricular septum; CM: compact myocardium; TM: trabecular myocardium. p Values were calculated using Student's t-test. (**c**) Immunoblot analysis of different tissues isolated from $Rit1^{+/+}$ (wild type), $Rit1^{+/-}$ (heterozygous), and $Rit1^{-/-}$ (knockout) adult mice. (**d**) Percentage of *Lztr1* genotypes obtained upon weaning (21 days of age) the offspring of *Lztr1* heterozygous mutant intercrosses in either a $Rit1^{+/+}$ (n = 19) or $Rit1^{-/-}$ (n = 85) background. All these mice were maintained in a C57BL/6 N background. (**e**) Histological analysis of the heart of *Lztr1; Rit1* double knockout E18.5 embryos (n = 3). (**f**) Immunoblot analysis of lysates isolated from primary MEF with the indicated genotypes. (**g**) Primary MEF derived from wild type, *Lztr1* knockout, *Rit1* knockout, and *Lztr1; Rit1* double knockout were starved overnight and stimulated with 10% FBS during the indicated times. Protein lysates were immunoblotted as indicated. Immunoblots represent a representative result from two biological replicates. (**h**) Quantitative PCR analysis of *Spry2* and *Dusp6* mRNA levels in primary MEF with the indicated genotypes stimulated for 1 hr with 10% FBS (n = 3 biological replicates). p Values were calculated using Student's t-test. p values: * (p < 0.05); ** (p < 0.01); *** (p < 0.005).

The online version of this article includes the following source data and figure supplement(s) for figure 4:

**Source data 1.** Raw data for *Figure 4* and *Figure 4—figure supplement 1*.

**Figure supplement 1.** Phenotyping results of Lztr1/Rit1 double knockout mice and embryos.

*Lztr1/Rit1* DKO mice is correlated to a rescue of MAPK pathway hyperactivation, we isolated primary mouse embryonic fibroblasts (MEF) from wild type, *Lztr1* KO, *Rit1* KO, and *Lztr1/Rit1* DKO embryos (*Figure 4f*). We then subjected these MEFs to FBS stimulation and observed a noticeable decrease in MAPK signaling in cells devoid of both LZTR1 and RIT1 compared to LZTR1 KO cells, both by immunoblot and using the two well-characterized MAPK-regulated transcriptional targets *Spry2* and *Dusp6* (*Figure 4g–h*). This indicated that, in our MEF cells, MAPK pathway hyperactivation is mediated by RIT1 protein stabilization in the absence of CRL3$^{LZTR1}$ regulation.

Many human RAS proteins are highly conserved in less complex organisms and a common ancestor Ras protein has been described in distant relatives such as yeast and slime mold, among other Eukarya (*Cox and Der, 2010*). Phylogenetic analyses of certain regulators of RAS function, such as GAPs and GEFs, highlight the prominence of early RAS activity regulation and its robust coevolution; with ancestral genomes harboring multiple GAP and GEF genes for a single RAS GTPase and a notable expansion of GAP genes congruent with the expansion of RAS GTPases in eukaryotes (*van Dam et al., 2011*; *van Dam et al., 2009*). The high percentage of GTP-loaded RIT1 in mammalian cells indicates the potential absence or evolutionary loss of GAP-mediated regulation and highlights the importance of alternative RIT1 regulatory mechanisms, such as protein turnover. Here, we show that in contrast to the highly conserved protein KRAS, both RIT1 and LZTR1 proteins are less conserved in lower organisms. Despite this, LZTR1 orthologs of invertebrate model organisms retain preferential binding toward their respective RIT1, but not RAS, orthologs. Moreover, fruit flies exhibit strong conservation of the LZTR1-mediated protein degradation mechanism with specificity toward RIT1, as we had previously described with human orthologs, but minimal activity toward other fruit fly RAS GTPases. These data indicate a functional co-evolution between LZTR1 and RIT1.

We also demonstrate that, in contrast to previous reports, LZTR1 is haplosufficient in mice (*Steklov et al., 2018*). However, homozygous knockout mutations are not tolerated due to embryonic lethality caused by aberrant cardiovascular development concomitant with severe peripheral hemorrhaging and ventricular septal defects, which can be rescued by *Rit1* germline deletion. Interestingly, Sewduth et al., demonstrate that pharmacological inhibition of the RAF/MEK/ERK MAPK or AKT pathway was unable to fully rescue the embryonic lethality and vascular defects of mice with conditional *Lztr1* KO in blood vessels (*Sewduth et al., 2020*). Given that our *Lztr1/Rit1* DKO mice did not exhibit cardiovascular abnormalities during embryogenesis, excessive accumulation of Rit1 protein in Lztr1 KO cells likely results in the hyperactivation or dysregulation of additional Rit1 effector pathways that contribute to *Lztr1$^{-/-}$* embryonic lethality (*Cuevas-Navarro et al., 2021*; *Meyer Zum Büschenfelde et al., 2018*; *Vichas et al., 2021*).

While other RAS GTPases have cognate GAPs and GEFs that regulate their activity, RIT1 GAPs and GEFs remain to be identified. Supported by the robust functional and evolutionary relationship

between LZTR1 and RIT1 described here, we posit that LZTR1 may have co-evolved with RIT1 as an alternative mechanism to regulate RIT1 GTPase activity.

# Materials and methods

## Key resources table

| Reagent type (species) or resource | Designation | Source or reference | Identifiers | Additional information |
|---|---|---|---|---|
| Strain, strain background (*Escherichia coli*) | BL21(DE3) | NEB | C2527H | |
| Genetic reagent (*D. melanogaster*) | Lztr1[1] | This paper | | CG3711/Lztr1 knockout |
| Genetic reagent (*D. melanogaster*) | Lztr1[2] | This paper | | CG3711/Lztr1 knockout |
| Genetic reagent (*D. melanogaster*) | Ras[HA] | This paper | | Transgene of HA-Ras at attP-86Fb landing site |
| Genetic reagent (*D. melanogaster*) | Ric[HA] | This paper | | Transgene of HA-Ric at attP-86Fb landing site |
| Genetic reagent (*M. musculus*) | Lztr1[-/-] | EUCOMM | Lztr1[tm1a(EUCOMM)Wtsi]; RRID: IMSR_EM:06794 | Lztr1 knockout |
| Genetic reagent (*M. musculus*) | Rit1[-/-] | This paper | | Rit1 knockout |
| Genetic reagent (*M. musculus*) | Lztr1[-/-];Rit1[-/-] | This paper | | Lztr1 and Rit1 double knockout |
| Genetic reagent (*M. musculus*) | 129S1/Svlmj | Jackson Laboratories | 002448; RRID:IMSR_JAX:002448 | |
| Recombinant DNA reagent | pcDNA3-DEST-Flag-SOS[cat] (*H. sapiens*) | This paper | | Vector to express Flag-SOS1 (residues 564–1049) in mammalian cells. |
| Recombinant DNA reagent | pcDNA3-DEST-HA-LZTR1 (*H. sapiens*) | This paper | | Vector to express HA-LZTR1 in mammalian cells. |
| Recombinant DNA reagent | pcDNA3-DEST-HA-LZTR1 (*M. musculus*) | This paper | | Vector to express HA-LZTR1 in mammalian cells. |
| Recombinant DNA reagent | pcDNA3-DEST-HA-LZTR1 (*D. rerio*) | This paper | | Vector to express HA-LZTR1 in mammalian cells. |
| Recombinant DNA reagent | pcDNA3-DEST-HA-LZTR1 (*D. melanogaster*) | This paper | | Vector to express HA-LZTR1 in mammalian cells. |
| Recombinant DNA reagent | pGEX-6P-DEST-KRAS (*H. sapiens*) | This paper | | Vector to express GST-KRAS in *E. coli* cells. |
| Recombinant DNA reagent | pGEX-6P-DEST-RIT1 (*H. sapiens*) | This paper | | Vector to express GST-RIT1 in *E. coli* cells. |
| Recombinant DNA reagent | pGEX-6P-DEST-KRAS (*M. musculus*) | This paper | | Vector to express GST-KRAS in *E. coli* cells. |
| Recombinant DNA reagent | pGEX-6P-DEST-RIT1 (*M. musculus*) | This paper | | Vector to express GST-RIT1 in *E. coli* cells. |
| Recombinant DNA reagent | pGEX-6P-DEST-KRAS (*D. rerio*) | This paper | | Vector to express GST-KRAS in *E. coli* cells. |
| Recombinant DNA reagent | pGEX-6P-DEST-RIT1 (*D. rerio*) | This paper | | Vector to express GST-RIT1 in *E. coli* cells. |
| Recombinant DNA reagent | pGEX-6P-DEST-RAS (*D. melanogaster*) | This paper | | Vector to express GST-KRAS in *E. coli* cells. |
| Recombinant DNA reagent | pGEX-6P-DEST-RIC (*D. melanogaster*) | This paper | | Vector to express GST-RIT1 in *E. coli* cells. |
| Antibody | Anti-HA (Rabbit monoclonal) | Cell Signalling Technology | Cat#: 3724; RRID: AB_1549585 | WB (1:3,000) |

*Continued on next page*

*Continued*

| Reagent type (species) or resource | Designation | Source or reference | Identifiers | Additional information |
|---|---|---|---|---|
| Antibody | Anti-Flag (Rabbit monoclonal) | Cell Signalling Technology | Cat#: 14793; RRID: AB_2572291 | WB (1:3,000) |
| Antibody | Anti-p-ERK1/2 (Rabbit monoclonal) | Cell Signalling Technology | Cat#: 4370; RRID: AB_2315112 | WB (1:1,000) |
| Antibody | Anti-ERK1/2 (Rabbit monoclonal) | Cell Signalling Technology | Cat#: 4696; RRID: AB_390780 | WB (1:2,000) |
| Antibody | Anti-p-MEK1/2 (Rabbit monoclonal) | Cell Signalling Technology | Cat#: 9154; RRID: AB_2138017 | WB (1:1,000) |
| Antibody | Anti-MEK1/2 (Mouse monoclonal) | Cell Signalling Technology | Cat#: 4694; RRID: AB_10695868 | WB (1:1,000) |
| Antibody | Anti-RIT1 (Rabbit polyclonal) | Abcam | Cat#: ab53720; RRID: AB_882379 | WB (1:1,000) |
| Antibody | Anti-b-Actin (Mouse monoclonal) | Sigma-Aldrich | Cat#: A2228; RRID: AB_476697 | WB (1:10,000) |
| Antibody | Anti-a-Tubulin (Mouse monoclonal) | Sigma-Aldrich | Cat#: T6199; RRID: AB_477583 | WB (1:10,000) |
| Antibody | Anti-KRAS (Mouse monoclonal) | Sigma-Aldrich | Cat#: WH0003845M1; RRID: AB_1842235 | WB (1:500) |
| Antibody | Anti-Ras (Rabbit monoclonal) | Cell Signalling Technology | Cat#: 4370; RRID: AB_2910195 | WB (1:1,000) |
| Antibody | Anti-NRAS (Mouse monoclonal) | Santa Cruz Biotechnology | Cat#: sc-31; RRID: AB_628041 | WB (1:1,000) |
| Antibody | Anti-HRAS (Rabbit polyclonal) | Santa Cruz Biotechnology | Cat#: sc-520; RRID: AB_631670 | WB (1:500) |
| Antibody | Anti-LZTR1 (Mouse monoclonal) | Santa Cruz Biotechnology | Cat#: sc-390166X; RRID: AB_2910196 | WB (1:1,000) |
| Sequence-based reagent | Dusp6_F | This paper | PCR primers | TCCTATCTCGGATCACTGGAG |
| Sequence-based reagent | Dusp6_R | This paper | PCR primers | GCTGATACCTGCCAAGCAAT |
| Sequence-based reagent | Spry2_F | This paper | PCR primers | CATCGCTGGAAGAAGAGGAT |
| Sequence-based reagent | Spry2_R | This paper | PCR primers | CATCAGGTCTTGGCAGTGT |
| Sequence-based reagent | Tbp_F | This paper | PCR primers | CCTTGTACCCTTCACCAATGAC |
| Sequence-based reagent | Tbp_R | This paper | PCR primers | ACAGCCAACATTCACGGTAGA |

## DNA constructs

Human, mouse (*Mus musculus*), zebrafish (*Danio rerio*), and fruit fly (*Drosophila melanogaster*) RIT1 and KRAS orthologs were synthesized as *E. coli* codon-optimized gene blocks (IDT) and cloned into pDONR221 using Gateway BP cloning. SOS$^{cat}$ was subcloned from human SOS1 (residues 564–1049) into pDONR221. Gateway LR reaction was used to generate a pGEX-6P destination vector for bacterial expression. Human, mouse (*Mus musculus*), zebrafish (*Danio rerio*), and fruit fly (*Drosophila melanogaster*) LZTR1 orthologs were synthesized as *H. sapiens* codon-optimized gene blocks (IDT) and cloned into pDONR221. Gateway LR reaction was used to generate either pcDNA3-HA or pcDNA3-Flag-tagged destination vectors for mammalian expression. All plasmids were verified by Sanger sequencing.

## Recombinant proteins

GST-tagged recombinant proteins were expressed in BL21 (DE3) *E. coli* cells and expression was induced with 0.2 mM IPTG for 14–16 hr at 18 °C. Cells were lysed by sonication in 50 mM Tris-HCl (pH 8.0), 300 mM NaCl, 5% glycerol, 1 mM DTT. Proteins were immobilized on Glutathione Sepharose 4B beads (Cytiva Life Sciences), washed extensively, and stored as a 50% glycerol bead suspension at −20 °C. HA- or Flag-tagged recombinant proteins were expressed in HEK-293T cells by transient transfection and collected in lysates after 24 hr.

## Immunoblot

Protein samples were prepared from frozen mouse tissues using RIPA buffer and a Dounce tissue homogenizer. *Drosophila* samples were prepared from either frozen L3 instar larvae, adults, or isolated fly heads and were lysed in RIPA lysis buffer with a Dounce homogenizer. Protein lysates were cleared by centrifugation. For immunoblot detection, samples were separated by SDS-PAGE (NuPAGE) and transferred onto nitrocellulose membranes using iBlot2. Membranes were blocked using 2.5% skimmed milk in TBS-T buffer for 1 hr at room temperature and incubated with appropriate primary antibodies overnight. Detection was performed using HRP-linked secondary antibodies and developed with Amersham ECL (Cytiva Life Sciences) and X-ray films.

Antibodies used in this work were: HA (Cell Signaling Technologies, Cat #3724; 1:3,000), β-Actin (Sigma-Aldrich, Cat #A2228; 1:10,000), α-Tubulin (Sigma-Aldrich, Cat #T6199; 1:10,000), LZTR1 (Santa Cruz Biotechnology, Cat #sc-390166X; 1:1,000), RIT1 (Abcam, Cat #ab53720; 1:1,000), p-ERK (Cell Signaling Technologies, Cat #4370; 1:1000), ERK1/2 (Cell Signaling Technologies, Cat #4696; 1:2000), p-MEK (Cell Signaling Technologies, Cat #9154; 1:1000), MEK1/2 (Cell Signaling Technologies, Cat #4694; 1:1000), panRAS (Cell Signaling Technologies, Cat # 67648; 1:1000), NRAS (Santa Cruz Biotechnology, Cat #sc-31; 1:1,000), HRAS (Santa Cruz Biotechnology, Cat #sc-520; 1:500), and KRAS (Sigma-Aldrich, Cat # WH0003845M1; 1:500).

## Mice

The *Lztr1* allele (Lztr1$^{tm1a(EUCOMM)Wtsi}$) was previously described (*Castel et al., 2019*). Briefly, frozen sperm was obtained from the Knockout Mouse Project and IVF was performed on C57BL/6NTac eggs. Heterozygous mice were maintained in C57BL/6NTac congenic background. Every six months, new C57BL/6NTac males (Taconic) were introduced to our colony to avoid genetic drift. Homozygous knockout *Lztr1* embryos (E18.5) were obtained from timed pregnancies using a heterozygous x heterozygous breeding scheme. 129S1/Svlmj mice were purchased at the Jackson Laboratories. Mixed C56BL/6 N-129S1 mice were obtained by crossing *Lztr1* heterozygous male C56BL/6 N mice with 129S1 females for the first 5 backcrossings. Then,129S1 males were introduced for Y-chromosome fixing. Knockout *Lztr1* mice were considered to be in a pure 129S1 background after at least eight backcrosses. *Rit1* knockout mice were generated by Cyagen using CRISP/Cas9-mediated large homology arm recombination in C57BL/6NTac fertilized eggs. Briefly, a cassette containing a loxP-STOP-loxP was engineered to be located at the first coding exon of *Rit1* and was microinjected with appropriate gRNA and Cas9 mRNA into zygotes and transferred into the oviducts of pseudopregnant females. Mice with germline transmission of the *Rit1* allele were used as founders to establish the *Rit1* knockout homozygous colony. This study was performed in strict accordance with the recommendations in the Guide for the Care and Use of Laboratory Animals of the National Institutes of Health. All of the animals were handled according to approved institutional animal care and use committee (IACUC) protocols (#AN165444 and #AN179937) of the University of California San Francisco.

## Mouse phenotyping

Noonan syndrome-like phenotypes in mice were analyzed as previously reported (*Castel et al., 2019*). Briefly, mice were weaned at 3 weeks of age, weighted at 4 weeks, skull CT was performed at 8 weeks and then mice were euthanized for heart measurements. Sample size was calculated based on anticipated values from our previous phenotyping experiments using a Rit1$^{M90I}$ Noonan syndrome mouse model (*Castel et al., 2019*). For skull morphometry, sample size was determined to be at least n = 5 to achieve an alpha of 0.05 with a power of 80%. For body weight, sample size was determined to be at least n = 13, and for organ-to-body weight measurements, sample size was determined to be at least n = 6, following the same statistical parameters described above.

For E18.5 embryos, trunks were fixed overnight in 10% buffered formalin, dehydrated with ethanol, and embedded in paraffin. For cardiac analysis, the whole trunk was sectioned transversely at 6 µm and H&E staining was performed following standard protocols at the Histology Core Facility of the National Center for Cardiovascular Research (CNIC). Ventricular wall measurements were obtained from H&E-stained sections. The length of at least 4 lines from a minimum of 8 sections obtained from 3 different embryos was measured with Ruler tool (NDP.View 2 Software). Statistical analyses were carried out using Prism 7 (GraphPad). Data are presented as mean ± SEM unless stated otherwise. Statistical significance was determined by performing 2-tailed, unpaired Student's *t* tests when

comparing 2 groups. p Values of less than 0.05 are considered significant. At least three independent dissections were performed to obtain E18.5 embryos.

## RT-qPCR

Total RNA from MEFs was isolated using the RNeasy kit (Qiagen) according to the manufacturer's instructions. cDNA was obtained by reverse transcription (RT) of 1 µg RNA using qScript XLT cDNA SuperMix (QuantaBio; 95161). Ten ng of cDNA was diluted in nuclease-free water and ran in technical triplicates using PowerUp SYBR Green Master Mix (Applied Biosystems) on a QuantStudio 5 (Thermo Fisher Scientific). Tbp (TATA-box binding protein) was used as an endogenous control.

Primer sequences were:

*Tbp* Fw: CCTTGTACCCTTCACCAATGAC; *Tbp* Rv: ACAGCCAACATTCACGGTAGA
*Spry2* Fw: CATCGCTGGAAGAAGAGGAT; *Spry2* Rv: CATCAGGTCTTGGCAGTGT
*Dusp6* Fw: TCCTATCTCGGATCACTGGAG; *Dusp6* Rv:GCTGATACCTGCCAAGCAAT

## *Drosophila* strains and crosses

Flies were maintained on standard food medium under a 12:12 light:dark cycle at 25 °C. The *Lztr1[1]* allele was generated by CRISPR-Cas9-mediated mutagenesis using transgenic *nanos-Cas9* (**Kondo and Ueda, 2013**; **Ren et al., 2013**) and guide RNA (gRNA) lines. Briefly, a double gRNA expression construct directed against the following protospacer sequences, 5´- GAAGCAAGCACACAGT GG-3' and 5´-GATGCGATGTTTGTATTCGG-3' (both corresponding to the upper strand), was generated in vector *pBFv-U6.2B* (**Kondo and Ueda, 2013**) and inserted via ΦC31 integrase-mediated transformation at the *attP40* landing site (**Bischof et al., 2007**). The resulting line was then crossed to *nanos-Cas9*-expressing flies to isolate Lztr1 mutations, including *Lztr1[1]*. *Lztr1[2]* was obtained via CRIS-PR-Cas9 engineering by GenetiVision Corporation and verified by PCR amplification and sequencing. HA-tagged Ric- and Ras-expressing constructs were ordered as custom genes from GenScript and transferred into the pattB vector (**Bischof et al., 2013**). Both contain the corresponding Ric and Ras genomic sequences including their respective 5' and 3' regulatory regions, with the HA coding sequence inserted after the initiation codon. Transgenic lines for each construct were established by ΦC31-mediated integration at the *attP-86Fb* landing site. Both homozygous lines are viable and fertile and were placed in the *Lztr1[2]* mutant background by standard crosses.

## Quantification of protein abundance by mass spectrometry

*Drosophila* head lysates (~300 µg in 100 µL per sample) were precipitated with acetone by adding 400 µL of ice-cold acetone, vortexing the lysates briefly, incubating the lysates at –20 °C for 1 hr, centrifuging at 15,000 x g for 10 min and decanting the supernatants. The protein pellets were then dissolved in 20 µL of 50 mM Tris-HCl buffer, pH 8.0 containing 8 M urea with vortexing. The samples were reduced by adding 1.2 µL of 100 mm dithiothreitol and vortexing for 30 min at room temperature (RT), followed by alkylation with added 3.5 µL of 100 mM iodoacetamide and gentle vortexing for 30 min in the dark for 30 min at RT. One µL from each sample was used to quantify protein amounts using BCA protein assay kit (Thermo Fisher Scientific, USA) according to the manufacturer's protocol. The samples were diluted by adding 90 µL of 50 mM Tris-HCl buffer, pH 8.0 containing 1 mM $CaCl_2$ and digested with 6 µg trypsin overnight at 37 °C with agitation. The samples were then acidified by adding 1 µL of formic acid, and the digested peptides were desalted using Sep-Pak C18 Classic Cartridges (Waters, USA) using the manufacturer's protocol.

For LC-MS/MS analysis, digested and desalted peptides were suspended in 0.1% formic acid, and ~0.5 ug was injected per sample on the EasySpray 50 cm C18 column (ES903, Thermo Fisher Scientific, USA) using Acquity UPLC M-Class System (Waters, USA) on line with Orbitrap Fusion Lumos Tribrid Mass Spectrometer (Thermo Fisher Scientific, USA). The column was held at 45 °C, and a 185 min low-pH reversed phase chromatography linear gradient was performed using a conventional two-buffer system (buffer A: 0.1% formic acid in MS-grade water; buffer B: 0.1% formic acid in acetonitrile) by increasing buffer B concentration from 3.5% to 30% over 185 min, followed by a 2 min wash to 50%, with a constant flow rate of 300 nL/min. The mass spectrometer was operated in positive ionization mode with the spray voltage of 2.5 KV, ion transfer tube temperature of 275 °C, RF lens at 30%, and internal calibration set to Easy-IC. MS1 spectra were collected in the Orbitrap in

profile mode at 120 K resolution, 375–1500 m/z mass range, 50ms maximum injection time (IT), and automatic gain control (AGC) target of 4.0e5. Precursor ions charged 2 + to 7 + with MS1 intensity above the threshold of 2.0e4 were isolated in the Quadruple using 1.6 Th isolation window, fragmented using higher-energy collisional dissociation (HCD) with 30% collision energy, and detected in the Orbitrap in Centroid mode at 30 K resolution, scan range set to Auto Normal, 100ms maximum IT, and 5.0e4 AGC target. The maximum cycle time was set to 3 s and dynamic exclusion to 30 s with +/-10 ppm tolerance.

MS.raw data files were converted to peak lists using PAVA in-house script and searched with Protein Prospector (v6.0.0) (*Chalkley et al., 2008*) against *Drosophila* UniProtKB FASTA-formatted database, which included proteins and their splicing isoforms downloaded on 2019-04-19 (uniprot.org) and a corresponding random-concatenated database of decoy peptides added with Protein Prospector. Instrument was set to "ESI-Q-high-res." Enzyme was set to Trypsin. Up to 2 missed cleavages were allowed and up to 2 post-translational modifications per peptide. "Carbamidomethyl (C)" was set as a constant modification, and default variable modifications were allowed: Acetyl (Protein N-term), Acetyl +Oxidation (Protein N-term M), Gln- > pyro Glu (N-term Q), Met-loss (Protein N-term M), Met-loss +Acetyl (Protein N-term M), Oxidation (M), and Oxidation (P). Precursor charges were set to 2 + to 5+, MS1 mass tolerance to 10 ppm, and MS2 mass tolerance to 20 ppm. FDR filters for peptides were set to 1% and for proteins to 5%. The search results were exported from Protein prospector as tab-delimited text and as.blib spectral library, which was imported into Skyline (v20) along with.raw files for quantification using MS1 filtering (*Schilling et al., 2012*). The identified peaks were detected by training built-in mProphet model against corresponding random-sequence decoy peptides (*Reiter et al., 2011*), peaks with an FDR < 1% were integrated. Built-in MSstats (v 3.13) (*Choi et al., 2014*) plugin in Skyline was used to normalize runs by median centering log2 precursor intensities, to calculate aggregate protein abundances, and to estimate their statistical significance.

## Acknowledgements

We thank Tony Huynh and Juan Antonio Camara Serrano for help with microCT imaging and Stephanie Mo for feedback on the manuscript. JG-B was funded by Programa "Atracción de Talento" de la Comunidad de Madrid. This work was supported by the NCI (1F31CA265066 to AC-N), (R35CA197709 to FM), (K99CA245122 to PC) and the Department of Defense Neurofibromatosis Research Program (W81XWH-20-1-0391 to PC). We thank the UCSF Mass Spectrometry Facility and A L Burlingame for providing MS instrumentation support for this project (funded by the NIH grants P41GM103481 and S10OD016229). GJ and LR-M were funded by grants from the Spanish Government (BFU2017-87244-P, PID2020-119248GB-I00 and Predoctoral contract BES-2015–071486).

## Additional information

### Competing interests

Frank McCormick: is a consultant for Ideaya Biosciences, Kura Oncology, Leidos Biomedical Research, Pfizer, Daiichi Sankyo, Amgen, PMV Pharma, OPNA-IO, and Quanta Therapeutics and has received research grants from Boehringer-Ingelheim and is a consultant for and cofounder of BridgeBio Pharma. Pau Castel: PC is a founder and advisory board of Venthera. The other authors declare that no competing interests exist.

### Funding

| Funder | Grant reference number | Author |
| --- | --- | --- |
| National Cancer Institute | F31CA265066 | Antonio Cuevas-Navarro |
| National Cancer Institute | R35CA197709 | Frank McCormick |
| National Cancer Institute | R00CA245122 | Pau Castel |
| DOD CDMRP Neurofibromatosis Research Program | W81XWH-20-1-0391 | Pau Castel |

| Funder | Grant reference number | Author |
|---|---|---|
| Comunidad de Madrid | Programa "Atracción de Talento" | Joaquim Grego-Bessa |

The funders had no role in study design, data collection and interpretation, or the decision to submit the work for publication.

## Author contributions

Antonio Cuevas-Navarro, Data curation, Investigation, Methodology, Visualization, Writing – original draft, Writing – review and editing; Laura Rodriguez-Muñoz, Katherine A Rauen, Investigation, Methodology, Writing – review and editing; Joaquim Grego-Bessa, Investigation, Methodology, Visualization, Writing – review and editing; Alice Cheng, Investigation, Writing – review and editing; Anatoly Urisman, Formal analysis, Investigation, Methodology, Writing – review and editing; Frank McCormick, Funding acquisition, Supervision, Writing – review and editing; Gerardo Jimenez, Funding acquisition, Investigation, Methodology, Writing – review and editing; Pau Castel, Conceptualization, Funding acquisition, Investigation, Methodology, Supervision, Visualization, Writing – original draft, Writing – review and editing

## Author ORCIDs

Joaquim Grego-Bessa (ID) http://orcid.org/0000-0002-0938-2346
Anatoly Urisman (ID) http://orcid.org/0000-0001-8364-5303
Pau Castel (ID) http://orcid.org/0000-0002-4972-4347

## Ethics

This study was performed in strict accordance with the recommendations in the Guide for the Care and Use of Laboratory Animals of the National Institutes of Health. All of the animals were handled according to approved institutional animal care and use committee (IACUC) protocols (#AN165444 and #AN179937) of the University of California San Francisco.

## Decision letter and Author response

Decision letter https://doi.org/10.7554/eLife.76495.sa1
Author response https://doi.org/10.7554/eLife.76495.sa2

# Additional files

## Supplementary files

• Transparent reporting form

## Data availability

All data generated or analysed during this study are included in the manuscript and supporting file; Source Data files have been provided for all Figures.

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
