## [Editor Report]

Using elegant cross-species biochemistry and genetic approaches, this paper describes the role of the ubiquitin adaptor protein LZTR1 in regulation of the RAS-related GTPase RIT1 as its principal substrate involved in the RASopathy, Noonan syndrome. Although this work does not fully rule out the involvement of canonical RAS isoforms in LZTR1-associated RASopathies in humans, the extensive genetic experiments in *Drosophila* and mouse presented here suggest that pathological phenotypes observed in LZTR1-linked RASopathy models are mediated primarily by its target RIT1, and not by canonical RAS isoforms.

---

## [Decision Letter]

**Decision letter after peer review:**

Thank you for submitting your article "Cross-species analysis of LZTR1 loss-of-function mutants demonstrates dependency to RIT1 orthologs" for consideration by *eLife*. Your article has been reviewed by 3 peer reviewers, including Alice Berger as Reviewing Editor and Reviewer #1, and the evaluation has been overseen by Jonathan Cooper as the Senior Editor. The following individual involved in review of your submission has agreed to reveal their identity: Daniel Abankwa (Reviewer #3).

Essential revisions:

1) The pulldown experiments in Figure 1b should state the number of times the experiments were each repeated. In addition, a positive control for Ras pulldown should be included to ensure that the GST-Ras fusions are properly folded and capable of interacting with known effectors or other binding proteins.

2) Figure 2a-b or accompanying text should state the number of flies analyzed of each genotype.

3) Figure 3a-d should state the number of mice of each genotype analyzed.

4) Please provide quantification of the rescue phenotype shown in Figure 4e across a population of mice of each genotype.

5) The weakest conclusion in the papers is that Lztr1 inactivation functions through upregulation of MAPK signaling and that this phenotype is reversed by Rit1 inactivation (Figure 4g). The difference in MAPK signaling between Lztr1-/-/Rit1-/- cells and Lztr1-/- cells is not clear. This claim needs to be bolstered by additional evidence. How many repeats were performed, this information is not in the legend? Importantly, data in Figure 4g show that LZTR1 levels are serum-induced and then remain increased long term (24h). Thus, LZTR1 levels peak later than pMEK levels. Surprisingly, the modulation of LZTR1 is not reflected in the RIT1 levels in these blots. Here it would be important to see how RAS (ideally Hras and Kras) levels are modulated. All claims need quantification in particular those relating to pERK differences, which are not as obvious as pMEK differences. Could it be that in this particular background RIT1-levels are relatively uncoupled from LZTR regulation (Figure 4g)? Why?

*Reviewer #1 (Recommendations for the authors):*

Experimental considerations:

1) The pulldown experiments in Figure 1b should state the number of times the experiments were each repeated. In addition, a positive control for Ras pulldown should be included to ensure that the GST-Ras fusions are properly folded and capable of interacting with known effectors or other binding proteins.

2) Figure 2a-b or accompanying text should state the number of flies analyzed of each genotype.

3) Figure 3a-d should state the number of mice of each genotype analyzed.

4) Please provide quantification of the rescue phenotype shown in Figure 4e across a population of mice of each genotype.

5) The value of the gNOMAD analysis is unclear. Please explain what the screenshot is showing, and/or summarize the data in a more clear way.

6) Please explicitly state the background of both parents in the cross shown in Figure 4d. Is this cross on a pure C57Bl/6 background?

7) The authors have the unique ability to determine RAS and RIT1 levels in cells/tissues from Noonan Syndrome patients with LZTR1 mutation (as shown in Castel et al., Science 2019). To what extent are Ras protein levels altered in those samples? Including this data would enhance the impact of their work.

8) The weakest conclusion in the papers is that Lztr1 inactivation functions through upregulation of MAPK signaling and that this phenotype is reversed by Rit1 inactivation (Figure 4g). This reviewer has trouble seeing the "noticeable" difference in MAPK signaling between Lztr1-/-/Rit1-/- cells and Lztr1-/- cells. This claim needs to be bolstered by additional evidence.

*Reviewer #3 (Recommendations for the authors):*

1.

a) Figure 1b: Reciprocal pull-down experiments i.e. pulling down with LZTR1, could further strengthen these data. Based on the interaction proteomics data from Steklov et al., it could have been more relevant to examine Hras or Nras in Figure 1b. However, Steklov et al., found also Kras as ubiquitination target of LZTR1. In the current study, Kras may have been used due to its species-wide conservation. However, the above-mentioned background in regards to the other Ras isoforms should be stated or discussed.

b) The abstract states a bit vaguely, 'biochemical dependency', which may be better expressed as 'preferred interaction with RIT1 than RAS orthologs' or similar.

c) Given the constitutive GTP-bound state of RIT1 it may not be surprising that its activity level is regulated 11-fold via LZTR1. While this compares to a mere 1.5-fold regulation of canonical RAS levels in *Drosophila*, the latter needs to be multiplied by the probably 100 to 1000 fold-change in affinity e.g. for the RBD of the effector Raf, resulting in a total of 150 or 1500-fold activity change (PMID: 7852367). These activity ranges should be the actual values that are considered in order to assess the end-result of the regulation by LZTR1. However, canonical Ras activity will always depend on the coincidence of high local GEF and low GAP activity, while RIT1 might be active 'globally', a constellation particularly advantageous across long distances (e.g. in axons). This RAS-family activity consteallation could be discussed/ considered in their data interpretation.

2.

a) In mice the RIT1 tissue expression levels vary greatly (Figure 3g). Does this inversely correlate with LZTR1 levels in these tissues? If not, it could be assumed that an additional tissue specific modifier of the LZTR1/RIT1 output is highly relevant. In this context, do the RIT1 expression levels somehow correlate with the tissue-specific severity of NS-like phenotypes that are observed in LZTR1 -/- mice?

b) RIT1 and RIT2 are known to be highly expressed during brain development and were implicated in particular in neurite outgrowth (in Colicelli J 2004, PMID: 15367757). The increasing relevance for brain development (or in general complexity also with respect to vasculature) somewhat correlates with the evolutionary pattern the authors describe, namely that a higher divergence for RIT1 and LZTR1 orthologs is seen in less complex organisms. Neurite outgrowth, vascular outgrowth and developmental cell migration appear to often utilise similar molecular machineries, which would all impact on characteristic NS phenotypes. Can the authors therefore speculate on the specific relevance of RIT1 for the observed embryonal bleeding phenotypes (Figure 4a) in LZTR1 ko mice? The point is that it is quite possible that a specific relevance of RIT1 for vascularisation is sufficient to cause the lethal phenotype.

Given the discrepancy with the results reported by Steklov et al., and the cross-breeding data presented here, it is also plausible that additional modifiers are strongly involved in this complex process. In this context, the authors may first wish to consider the increased phenotypic relevance of the LZTR1/RIT1 coupling from fly (hardly any wing vein defects) to mouse (embryonal lethality)? This could be speculated on in the discussion.

c) The rescue in the double-knockout mice may be mostly an outcome of the rescue of this particular vascularization phenotype. It is not entirely clear, whether all LZTR1-/- associated defects are restored, data are mentioned but not shown (L. 264). The authors should provide the data or change their statement. They are encouraged furthermore to discuss the above points and eventually adjust their concluding statements.

d) It could have been interesting to examine the RIT1 and RAS levels in the heterozygous LZTR1 knockout animals from Steklov and see, whether they are increased; yet it is understood this is too much extra work. However, Steklov find Hras and Nras as main interactors of LZTR1, while here typically pan-Ras (unclear in Figure 3g) or Kras (Figure 1b) were considered. Data in Figure 3g need to be quantified, so that the (pan-)Ras effect is clearer. In the end, the fact that Ras is mildly up-modulated across all tissues may be more consistent with the multi-system defects observed in RASopathies.

One may wish to modify the statement that RIT1 is 'the' critical substrate of LZTR1 (L. 261), but rather 'an important'. Furthermore, it should be emphasized that only a 'partial' rescue (L.260) is observed

3.

a) Data in Figure 4g need to be quantified and sufficiently reproduced (i.e. in some repeats). How many repeats were performed, this information is not in the legend? Importantly, data in Figure 4g show that LZTR1 levels are serum-induced and then remain increased long term (24h). Thus, LZTR1 levels peak later than pMEK levels. Surprisingly, the modulation of LZTR1 is not reflected in the RIT1 levels in these blots. Here it would be important to see how RAS (ideally Hras and Kras) levels are modulated. All claims need quantification in particular those relating to pERK differences, which are not as obvious as pMEK differences. Could it be that in this particular background RIT1-levels are relatively uncoupled from LZTR regulation (Figure 4g)? Why?

b) If RIT1 is the major target responsible for the LZTR1 ko phenotype then how can a MEKi rescue the lethal phenotype as reported by Steklov et al.? Does RIT1 also bind to canonical RAS effectors? How are LZTR1 and RIT1 levels modulated with a MEKi? While such MEKi experiments during pregnancy are too laborious, it could be possible to test effects using MEFs such as in Figure 4g.

---

## [Author Response]

Essential revisions:1) The pulldown experiments in Figure 1b should state the number of times the experiments were each repeated. In addition, a positive control for Ras pulldown should be included to ensure that the GST-Ras fusions are properly folded and capable of interacting with known effectors or other binding proteins.

We have now indicated in the figure legend that the pulldown experiment in Figure 1b was repeated 3 times using two different batches of purified GST-tagged protein. Regarding the positive control for Ras interaction, this is a challenging experiment because the known effectors of Ras only bind in the GTP-bound conformation. In Figure 1b, the GST-fused recombinant Ras GTPases used for the pulldown assays are in the GDP-bound conformation, because we had previously described that only the RIT1 GDP-bound conformer binds LZTR1 (Castel et al., 2019). Therefore, interaction of RIT1 to LZTR1 cannot be compared to a classic effector (i.e. Raf RBD). Bacterial expression of Ras proteins has been extensively shown to result in enzymatically competent, properly folded, and GDP-bound GTPases (Lacal et al., 1984; Fang et al., 2016). In fact, we routinely use these recombinant proteins for nucleotide exchange, an additional proof that these enzymes are properly folded.

Nevertheless, to try to address this important comment by the reviewers, we have decided to test the interaction with a catalytic fragment of SOS1 (SOS^cat^; residues 550 to 1050) that is sufficient for Ras-specific nucleotide exchange activity. SOS^cat^ contains the Cdc25 domain, which was previously demonstrated to bind RAS, distort the nucleotide binding site, and promote the release of either GDP or GTP (Boriack-Sjodin et al., 1998; Margarit et al., 2003). Given the highly conserved nature of the Cdc25 domain, we used the human protein to assess the structural integrity of all the RAS orthologs. As expected, FLAG-tagged SOS^cat^ was able to bind all the recombinant RAS, but not RIT1, orthologs. This data has now been included as Figure 1 – supplement 1.

2) Figure 2a-b or accompanying text should state the number of flies analyzed of each genotype.

We have now indicated in the figure legend that we scored 209 flies in this experiment.

3) Figure 3a-d should state the number of mice of each genotype analyzed.

We have now indicated in the figure legend the number of mice analyzed in each experiment.

4) Please provide quantification of the rescue phenotype shown in Figure 4e across a population of mice of each genotype.

We have now included the quantification of compact and trabecular myocardial thickness in *Lztr1*^-/-^ and *Lztr1*^-/^;*Rit1*^-/-^ embryos shown in Figure 4e.

In addition, based on the comments from multiple reviewers, we have now included an analysis of Noonan syndrome-like traits in the *Lztr1*^-/-^;*Rit1*^-/-^ double knockout adult mice (Figure 4-supplement 1). In these mice, we measured body weight, craniofacial dysmorphia by micoCT, and heart and spleen weight. As previously indicated, these mice did not display any noticeable phenotype.

5) The weakest conclusion in the papers is that Lztr1 inactivation functions through upregulation of MAPK signaling and that this phenotype is reversed by Rit1 inactivation (Figure 4g). The difference in MAPK signaling between Lztr1-/-/Rit1-/- cells and Lztr1-/- cells is not clear. This claim needs to be bolstered by additional evidence. How many repeats were performed, this information is not in the legend? Importantly, data in Figure 4g show that LZTR1 levels are serum-induced and then remain increased long term (24h). Thus, LZTR1 levels peak later than pMEK levels. Surprisingly, the modulation of LZTR1 is not reflected in the RIT1 levels in these blots. Here it would be important to see how RAS (ideally Hras and Kras) levels are modulated. All claims need quantification in particular those relating to pERK differences, which are not as obvious as pMEK differences. Could it be that in this particular background RIT1-levels are relatively uncoupled from LZTR regulation (Figure 4g)? Why?

This is a very important point. We agree that the differences in MAPK are mild; this is generally the case in Noonan syndrome alleles in which the dysregulation of the MAPK pathway is mild and differences can only be seen upon stimulation with growth factors. Due to this small increase in MAPK activation in the Lztr1^-/-^ cells, it is also difficult to appreciate the rescue in the Lztr1^-/-^/Rit1^-/-^ cells. Given that immunoblot is a semiquantitative assay, we have now undertaken quantitative PCR of known and well-characterized MAPK target genes (*Spry2* and *Dusp6*) (Ekerot et al., 2008; Ozaki et al., 2001; Wagle et al., 2018). Using this assay, we can conclude that Lztr1^/-^/Rit1^-/-^ cells rescue the increased MAPK activity in response to growth factor stimulation (Figure 4h). We have included this new data and included the number of repeats in the figure legend for both immunoblot and qPCR as requested.

We have previously observed changes in LZTR1 protein levels upon serum stimulation in other cell types and we are currently following up on this interesting observation. However, these changes in LZTR1 levels are not associated with changes in RIT1 protein levels. This is consistent with our unpublished observations that low levels of LZTR1 do not necessarily result in higher RIT1 levels (as assessed by knocking down LZTR1 with different degrees of efficacy), while complete knockout of LZTR1 results in RIT1 accumulation. We think this is explained by the fact that LZTR1-mediated degradation of RIT1 is extremely efficient and even in the presence of low levels of LZTR1, there is still proficient RIT1 proteolysis. These observations are in line with the haplosufficiency seen in mice and humans for NS phenotypes. Moreover, it is likely that RIT1 ubiquitination by CRL3^LZTR1^ requires a molecular trigger that is yet to be discovered; therefore, levels of LZTR1 are not necessarily a good readout of RIT1 levels.

Reviewer #1 (Recommendations for the authors):Experimental considerations:1) The pulldown experiments in Figure 1b should state the number of times the experiments were each repeated. In addition, a positive control for Ras pulldown should be included to ensure that the GST-Ras fusions are properly folded and capable of interacting with known effectors or other binding proteins.

We have now included the number of times the Figure 1b experiment has been repeated. We have also included the data from an additional repeat in which we have included a positive control. Note that the binding between LZTR1 and RIT1 is GDP-dependent, so these recombinant proteins are not expected to bind to known effectors such as Raf kinases (GTP-dependent interaction). Therefore, we have used SOS1 catalytic domain (SOS^cat^), which is highly conserved and can bind GDP-bound RAS. GST-RAS recombinant proteins are well-documented to fold properly and are purified in their GDP-bound form.

2) Figure 2a-b or accompanying text should state the number of flies analyzed of each genotype.

We have now included the number of flies analyzed in these experiments.

3) Figure 3a-d should state the number of mice of each genotype analyzed.

We have now included the number of mice analyzed in these experiments.

4) Please provide quantification of the rescue phenotype shown in Figure 4e across a population of mice of each genotype.

We have now included the quantification for the cardiovascular rescue.

5) The value of the gNOMAD analysis is unclear. Please explain what the screenshot is showing, and/or summarize the data in a more clear way.

We have clarified the results from the gNOMAD analysis in the main text and have removed the supplementary figure containing the screenshot to avoid confusion (we came to the conclusion that the screenshot does not add any additional valuable information).

6) Please explicitly state the background of both parents in the cross shown in Figure 4d. Is this cross on a pure C57Bl/6 background?

Yes, this experiment was carried out in the same background (C57BL/6N). We have now clarified this in the figure legend.

7) The authors have the unique ability to determine RAS and RIT1 levels in cells/tissues from Noonan Syndrome patients with LZTR1 mutation (as shown in Castel et al., Science 2019). To what extent are Ras protein levels altered in those samples? Including this data would enhance the impact of their work.

This is a great point. We have analyzed these samples in the past and found that RAS levels inconsistently change between the different patient samples. This is also the case in other fibroblast lines derived from additional Noonan syndrome families (for which we do not have matched parents and/or unaffected sibling samples). However, because these variants (SNV, splice, early stop codon) are not necessarily equivalent to the null mutations characterized in our mouse and fruit fly models, we think it is better to not include this data within the manuscript as it can lead to confusion.

8) The weakest conclusion in the papers is that Lztr1 inactivation functions through upregulation of MAPK signaling and that this phenotype is reversed by Rit1 inactivation (Figure 4g). This reviewer has trouble seeing the "noticeable" difference in MAPK signaling between Lztr1-/-/Rit1-/- cells and Lztr1-/- cells. This claim needs to be bolstered by additional evidence.

We agree that the changes in MAPK signaling are small and difficult to notice when analyzed by immunoblot. This is very common given that the Noonan syndrome alleles weakly activate the signaling and western blot is not a very quantitative technique. To address this point, we have now undertaken transcriptional analysis by quantitative PCR of two well-known MAPK-regulated genes, *Spry2* and *Dusp6*. We now show in a more quantitative manner that deletion of *Rit1* decreases the degree of MAPK activation in *Lztr1*^-/-^ MEFs (new Figure 4h).

Reviewer #3 (Recommendations for the authors):1.a) Figure 1b: Reciprocal pull-down experiments i.e. pulling down with LZTR1, could further strengthen these data. Based on the interaction proteomics data from Steklov et al., it could have been more relevant to examine Hras or Nras in Figure 1b. However, Steklov et al., found also Kras as ubiquitination target of LZTR1. In the current study, Kras may have been used due to its species-wide conservation. However, the above-mentioned background in regards to the other Ras isoforms should be stated or discussed.

We thank the reviewer for these suggestions. As the reviewer points out, the reason we decided to use KRAS for our analysis and in vitro experiments is because *Drosophila Ras1* (the only RAS gene in this organism) is more similar to human *KRAS* than to *NRAS* or *HRAS*. In our previous publication (Castel et al., 2019 Science), using pulldown assays, we did not see measurable LZTR1 binding to any of the classical RAS proteins KRAS, NRAS or HRAS.

Regarding reciprocal pull downs, we have attempted this in the past, but have not been very successful. We hypothesize that because the N-terminal tag is in the Kelch repeats (domain required for RIT1 interaction), antibodies used to pull down LZTR1 compete with RIT1 interaction.

b) The abstract states a bit vaguely, 'biochemical dependency', which may be better expressed as 'preferred interaction with RIT1 than RAS orthologs' or similar.

We have now changed this statement as recommended by the reviewer.

c) Given the constitutive GTP-bound state of RIT1 it may not be surprising that its activity level is regulated 11-fold via LZTR1. While this compares to a mere 1.5-fold regulation of canonical RAS levels in *Drosophila*, the latter needs to be multiplied by the probably 100 to 1000 fold-change in affinity e.g. for the RBD of the effector Raf, resulting in a total of 150 or 1500-fold activity change (PMID: 7852367). These activity ranges should be the actual values that are considered in order to assess the end-result of the regulation by LZTR1. However, canonical Ras activity will always depend on the coincidence of high local GEF and low GAP activity, while RIT1 might be active 'globally', a constellation particularly advantageous across long distances (e.g. in axons). This RAS-family activity consteallation could be discussed/ considered in their data interpretation.

This is an important point and we would like to clarify. The interaction between LZTR1 and RIT1 is GDPdependent, as we had previously described (Castel et al., 2019 Science). Recombinant RIT1 produced in bacteria is found in the GDP-loaded state; it is in mammalian cells where we have previously found high GTP loading of RIT1 (even upon growth factor starvation). We attribute this observation to the selective degradation of the GDP bound form by CRL3^LZTR1^.

2.a) In mice the RIT1 tissue expression levels vary greatly (Figure 3g). Does this inversely correlate with LZTR1 levels in these tissues? If not, it could be assumed that an additional tissue specific modifier of the LZTR1/RIT1 output is highly relevant. In this context, do the RIT1 expression levels somehow correlate with the tissue-specific severity of NS-like phenotypes that are observed in LZTR1 -/- mice?

This is a very interesting question, but unfortunately, we are unable to measure the levels in a panel of mouse tissues. While the RIT1 antibody works well for this application, in our hands, the LZTR1 antibody immunoreacts nonspecifically in many tissues, which makes it impossible to assess the levels.

b) RIT1 and RIT2 are known to be highly expressed during brain development and were implicated in particular in neurite outgrowth (in Colicelli J 2004, PMID: 15367757). The increasing relevance for brain development (or in general complexity also with respect to vasculature) somewhat correlates with the evolutionary pattern the authors describe, namely that a higher divergence for RIT1 and LZTR1 orthologs is seen in less complex organisms. Neurite outgrowth, vascular outgrowth and developmental cell migration appear to often utilise similar molecular machineries, which would all impact on characteristic NS phenotypes. Can the authors therefore speculate on the specific relevance of RIT1 for the observed embryonal bleeding phenotypes (Figure 4a) in LZTR1 ko mice? The point is that it is quite possible that a specific relevance of RIT1 for vascularisation is sufficient to cause the lethal phenotype.Given the discrepancy with the results reported by Steklov et al., and the cross-breeding data presented here, it is also plausible that additional modifiers are strongly involved in this complex process. In this context, the authors may first wish to consider the increased phenotypic relevance of the LZTR1/RIT1 coupling from fly (hardly any wing vein defects) to mouse (embryonal lethality)? This could be speculated on in the discussion.

This is an interesting point. (Meyer Zum Büschenfelde et al., 2018) have demonstrated that RIT1 can regulate cell migration and actin dynamics though an association with the Rho family GTPases, CDC42 and Rac1. These GTPases engage with cellular developmental programs (such as planar cell polarity) that coordinate the migration and organization of tissues during development, including the formation of vascular networks and neurite migration. One can speculate in higher organisms, upregulation of RIT1 protein levels during development may have profound effects on the organization of certain tissues, such as the vasculature, potentially through MAPK-independent mechanisms, such as the CDC41/Rac1 signaling node.

The difference in the phenotypic relevance of LZTR1/RIT1 between mice and flies could be attributed to differences in the spatial and temporal expression of RIT1 or LZTR1 (as well as the expression and/or existence of different downstream effector pathways regulated by RIT1) during the development of these two organisms.

c) The rescue in the double-knockout mice may be mostly an outcome of the rescue of this particular vascularization phenotype. It is not entirely clear, whether all LZTR1-/- associated defects are restored, data are mentioned but not shown (L. 264). The authors should provide the data or change their statement. They are encouraged furthermore to discuss the above points and eventually adjust their concluding statements.

This is an important point. We think that the rescue is due to the improvement of both the cardiac and the vascular phenotype. As shown in Figure 4e, in E18.5 embryos the cardiac phenotype is completely rescued. Unfortunately, we have not quantified the vascular phenotype, but we did not find any double knockout embryo with hemorrhages.

To further characterize the rescued phenotype of the double knockout animals, we have now included the data of our morphometric analysis of adult animals for NS-like phenotype. We did not find any noticeable differences in body weight, skull morphology, or heart and spleen weight (Figure 4 – supplement 1). In the future, it would be interesting to further phenotype these mice (especially when aged) to see if there are any other phenotypes that could be linked to LZTR1 loss independently of RIT1.

d) It could have been interesting to examine the RIT1 and RAS levels in the heterozygous LZTR1 knockout animals from Steklov and see, whether they are increased; yet it is understood this is too much extra work. However, Steklov find Hras and Nras as main interactors of LZTR1, while here typically pan-Ras (unclear in Figure 3g) or Kras (Figure 1b) were considered. Data in Figure 3g need to be quantified, so that the (pan-)Ras effect is clearer. In the end, the fact that Ras is mildly up-modulated across all tissues may be more consistent with the multi-system defects observed in RASopathies.One may wish to modify the statement that RIT1 is 'the' critical substrate of LZTR1 (L. 261), but rather 'an important'. Furthermore, it should be emphasized that only a 'partial' rescue (L.260) is observed

We have now included immunoblot analysis of RAS and RIT1 proteins in liver and lung protein extracts from Lztr1 heterozygous mice (C57BL6/N background). These mice were originated in the Knockout Mouse Project (KOMP) and are the same line used by (Steklov et al., 2018). As seen in these tissues, there is no consistent upregulation of any RAS protein in the Lztr1 heterozygous mice. Unfortunately, we were unable to assess other tissues, because the LZTR1 antibody shows extensive immunoreactivity in mouse lysates. This data has now included as Figure 3 – supplement 1e.

3.a) Data in Figure 4g need to be quantified and sufficiently reproduced (i.e. in some repeats). How many repeats were performed, this information is not in the legend? Importantly, data in Figure 4g show that LZTR1 levels are serum-induced and then remain increased long term (24h). Thus, LZTR1 levels peak later than pMEK levels. Surprisingly, the modulation of LZTR1 is not reflected in the RIT1 levels in these blots. Here it would be important to see how RAS (ideally Hras and Kras) levels are modulated. All claims need quantification in particular those relating to pERK differences, which are not as obvious as pMEK differences. Could it be that in this particular background RIT1-levels are relatively uncoupled from LZTR regulation (Figure 4g)? Why?

We have now included in the figure legend the number of repeats for experiment 4g.

As the reviewer has noticed, we have previously observed that LZTR1 levels can be modulated by FBS and we are currently following up on this interesting observation. These changes in LZTR1 protein levels are not associated with changes in RIT1 levels. This is consistent with our unpublished observations that low levels of LZTR1 do not necessarily result in higher RIT1 levels (as assessed by knocking down LZTR1 with different degrees of efficacy), while complete knockout of LZTR1 results in RIT1 accumulation. We think this is explained by the fact that LZTR1-mediated degradation of RIT1 is extremely efficient and even in the presence of low levels of LZTR1, there is still proficient RIT1 proteolysis. These observations are in line with the haplosufficiency seen in mice and humans for Noonan syndrome-related phenotypes. Moreover, it is likely that RIT1 ubiquitination by CRL3^LZTR1^ requires a molecular trigger that is yet to be discovered; therefore, levels of LZTR1 are not necessarily a good readout of RIT1 protein levels.

Because western blot is a semiquantitative technique, we have now included the results from our experiments in these cells using qPCR of well-known transcriptional targets of the MAPK (*Dusp6* and *Spry2*). We now clearly show that double knockout cells decrease the levels of MAPK activity seen in LZTR1 knockout cells (Figure 4e).

b) If RIT1 is the major target responsible for the LZTR1 ko phenotype then how can a MEKi rescue the lethal phenotype as reported by Steklov et al.? Does RIT1 also bind to canonical RAS effectors? How are LZTR1 and RIT1 levels modulated with a MEKi? While such MEKi experiments during pregnancy are too laborious, it could be possible to test effects using MEFs such as in Figure 4g.

RIT1 has been shown by us and others to activate MAPK (Berger et al., 2014; Castel et al., 2019; Fang et al., 2016, p. 1; Lo et al., 2021; Van et al., 2020, p. 1). In fact, RIT1 is one of the most common mutations in Noonan syndrome, which is a disorder driven by increased MAPK response (Aoki et al., 2013). Consistent with this, a few case reports have shown that the cardiac phenotype of RIT1 mutant Noonan syndrome ameliorated upon pharmacological treatment with a MEK1/2 allosteric inhibitor (Andelfinger et al., 2019). A plausible explanation of Steklov’s data is that the embryonic phenotype of LZTR1 knockout mice is due to RIT1-driven MAPK activation. This would be consistent with the fact that in our hands, we rescue this phenotype with the RIT1 knockout allele.

We are currently working on a project that explores the mechanism by which RIT1 can activate the MAPK; our data so far indicates that RIT1 can bind directly Raf (with low affinity), but is likely to still require RAS activity to activate MAPK.

References

Andelfinger G, Marquis C, Raboisson M-J, Théoret Y, Waldmüller S, Wiegand G, Gelb BD, Zenker M, Delrue M-A, Hofbeck M. 2019. Hypertrophic Cardiomyopathy in Noonan Syndrome Treated by MEK-Inhibition. Journal of the American College of Cardiology 73:2237–2239. doi:10.1016/j.jacc.2019.01.066

Aoki Y, Niihori T, Banjo T, Okamoto N, Mizuno S, Kurosawa K, Ogata T, Takada F, Yano M, Ando T, Hoshika T, Barnett C, Ohashi H, Kawame H, Hasegawa T, Okutani T, Nagashima Tatsuo, Hasegawa S, Funayama R, Nagashima Takeshi, Nakayama K, Inoue S-I, Watanabe Y, Ogura T, Matsubara Y. 2013. Gain-of-function mutations in RIT1 cause Noonan syndrome, a RAS/MAPK pathway syndrome. Am J Hum Genet 93:173–180. doi:10.1016/j.ajhg.2013.05.021

Berger AH, Imielinski M, Duke F, Wala J, Kaplan N, Shi G-X, Andres DA, Meyerson M. 2014. Oncogenic RIT1 mutations in lung adenocarcinoma. Oncogene 33:4418–4423. doi:10.1038/onc.2013.581

Boriack-Sjodin PA, Margarit SM, Bar-Sagi D, Kuriyan J. 1998. The structural basis of the activation of Ras by Sos. Nature 394:337–343. doi:10.1038/28548

Castel P, Cheng A, Cuevas-Navarro A, Everman DB, Papageorge AG, Simanshu DK, Tankka A, Galeas J, Urisman A, McCormick F. 2019. RIT1 oncoproteins escape LZTR1-mediated proteolysis. Science 363:1226–1230.

doi:10.1126/science.aav1444

Ekerot M, Stavridis MP, Delavaine L, Mitchell MP, Staples C, Owens DM, Keenan ID, Dickinson RJ, Storey KG, Keyse SM. 2008. Negative-feedback regulation of FGF signalling by DUSP6/MKP-3 is driven by ERK1/2 and mediated by Ets factor binding to a conserved site within the DUSP6/MKP-3 gene promoter. Biochem J 412:287–298. doi:10.1042/BJ20071512

Fang Z, Marshall CB, Yin JC, Mazhab-Jafari MT, Gasmi-Seabrook GMC, Smith MJ, Nishikawa T, Xu Y, Neel BG, Ikura M. 2016. Biochemical Classification of Disease-associated Mutants of RAS-like Protein Expressed in Many Tissues (RIT1). J Biol Chem 291:15641–15652. doi:10.1074/jbc.M116.714196

Lacal JC, Santos E, Notario V, Barbacid M, Yamazaki S, Kung H, Seamans C, McAndrew S, Crowl R. 1984. Expression of normal and transforming H-ras genes in *Escherichia coli* and purification of their encoded p21 proteins. Proc Natl Acad Sci U S A 81:5305–5309. doi:10.1073/pnas.81.17.5305

Lo A, Holmes K, Kamlapurkar S, Mundt F, Moorthi S, Fung I, Fereshetian S, Watson J, Carr SA, Mertins P, Berger AH. 2021. Multiomic characterization of oncogenic signaling mediated by wild-type and mutant RIT1. Sci Signal 14:eabc4520. doi:10.1126/scisignal.abc4520

Margarit SM, Sondermann H, Hall BE, Nagar B, Hoelz A, Pirruccello M, Bar-Sagi D, Kuriyan J. 2003. Structural evidence for feedback activation by Ras.GTP of the Ras-specific nucleotide exchange factor SOS. Cell 112:685–695.

doi:10.1016/s0092-8674(03)00149-1

Meyer Zum Büschenfelde U, Brandenstein LI, von Elsner L, Flato K, Holling T, Zenker M, Rosenberger G, Kutsche K. 2018. RIT1 controls actin dynamics via complex formation with RAC1/CDC42 and PAK1. PLoS Genet 14:e1007370. doi:10.1371/journal.pgen.1007370

Ozaki K, Kadomoto R, Asato K, Tanimura S, Itoh N, Kohno M. 2001. ERK pathway positively regulates the expression of Sprouty genes. Biochem Biophys Res Commun 285:1084–1088. doi:10.1006/bbrc.2001.5295

Steklov M, Pandolfi S, Baietti MF, Batiuk A, Carai P, Najm P, Zhang M, Jang H, Renzi F, Cai Y, Abbasi Asbagh L, Pastor T, De Troyer M, Simicek M, Radaelli E, Brems H, Legius E, Tavernier J, Gevaert K, Impens F, Messiaen L, Nussinov R, Heymans S, Eyckerman S, Sablina AA. 2018. Mutations in LZTR1 drive human disease by dysregulating RAS ubiquitination. Science 362:1177–1182. doi:10.1126/science.aap7607

Van R, Cuevas-Navarro A, Castel P, McCormick F. 2020. The molecular functions of RIT1 and its contribution to human disease. Biochem J 477:2755–2770. doi:10.1042/BCJ20200442

Wagle M-C, Kirouac D, Klijn C, Liu B, Mahajan S, Junttila M, Moffat J, Merchant M, Huw L, Wongchenko M, Okrah K, Srinivasan S, Mounir Z, Sumiyoshi T, Haverty PM, Yauch RL, Yan Y, Kabbarah O, Hampton G, Amler L, Ramanujan S, Lackner MR, Huang S-MA. 2018. A transcriptional MAPK Pathway Activity Score (MPAS) is a clinically relevant biomarker in multiple cancer types. NPJ Precis Oncol 2:7. doi:10.1038/s41698-018-0051-4